# Evaluation of Histone Demethylase Inhibitor ML324 and Acyclovir against *Cyprinid herpesvirus 3* Infection

**DOI:** 10.3390/v15010163

**Published:** 2023-01-05

**Authors:** Shelby Matsuoka, Gloria Petri, Kristen Larson, Alexandra Behnke, Xisheng Wang, Muhui Peng, Sean Spagnoli, Christiane Lohr, Ruth Milston-Clements, Konstantin Divilov, Ling Jin

**Affiliations:** 1Department of Biomedical Sciences, College of Veterinary Medicine, Oregon State University, Corvallis, OR 97331, USA; 2Department of Microbiology, College of Science, Oregon State University, Corvallis, OR 97331, USA; 3Department of Fisheries, Wildlife, and Conservation Sciences, Oregon State University, Newport, OR 97365, USA

**Keywords:** CyHV-3, temperature stress, histone demethylase inhibitors, ML324, acyclovir, real-time PCR, histopathology

## Abstract

*Cyprinid herpesvirus 3* (CyHV-3) can cause severe disease in koi and common carp (*Cyprinus carpio*). Currently, no effective treatment is available against CyHV-3 infection in koi. Both LSD1 and JMJD2 are histone demethylases (HD) and are critical for immediate-early (IE) gene activation essential for lytic herpesvirus replication. OG-L002 and ML324 are newly discovered specific inhibitors of LSD1 and JMJD2, respectively. Here, HD inhibitors were compared with acyclovir (ACV) against CyHV-3 infection in vitro and in vivo. ML324, at 20–50 µM, can completely block ~1 × 10^3^ PFU CyHV-3 replication in vitro, while OG-L002 at 20 µM and 50 µM can produce 96% and 98% inhibition, respectively. Only about 94% inhibition of ~1 × 10^3^ PFU CyHV-3 replication was observed in cells treated with ACV at 50 µM. As expected, CyHV-3 IE gene transcription of ORF139 and ORF155 was blocked within 72 h post-infection (hpi) in the presence of 20 µM ML324. No detectable cytotoxicity was observed in KF-1 or CCB cells treated for 24 h with 1 to 50 µM ML324. A significant reduction of CyHV-3 replication was observed in ~6-month-old infected koi treated with 20 µM ML324 in an immersion bath for 3–4 h at 1-, 3-, and 5-days post-infection compared to the control and ACV treatments. Under heat stress, 50–70% of 3–4-month-old koi survived CyHV-3 infection when they were treated daily with 20 µM ML324 in an immersion bath for 3–4 h within the first 5 d post-infection (dpi), compared to 11–19% and 22–27% of koi in the control and ACV treatments, respectively. Our study demonstrates that ML324 has the potential to be used against CyHV-3 infection in koi.

## 1. Introduction

*Cyprinid herpesvirus 3* (CyHV-3), commonly known as koi herpesvirus (KHV), is a pathogen of koi and common carp (*Cyprinus carpio*) that causes significant disease and mortality in young fry under 4 weeks old [1,2]. The mortality of juveniles from CyHV-3 infection can reach 100% within 2–3 weeks after infection [2,3]. The clinical signs of acute CyHV-3 infection include red and white mottled gills, gill hemorrhage, sunken eyes, pale patches or blisters on the skin, and external hemorrhages [1,2,3]. The virus can also infect the kidney, spleen, fin, intestine, and brain [4,5]. CyHV-3 is a member of the family *Alloherpesviridae* within the order of *Herpesvirales* [6]. Akin to herpesviruses from *Herpesviridae*, CyHV-3 also becomes latent in fish recovered from active or asymptomatic infections [7,8,9]. Therefore, CyHV-3 can be detected in healthy-looking koi or common carp [9,10]. CyHV-3 reactivation from latency can cause inflammation in various organs and death, and infectious viruses from reactivation can spread to naïve koi and cause disease [11,12]. No treatment or effective vaccine has been developed to control CyHV-3 infection in koi, one of the most popular ornamental fish.

Herpesvirus infections, such as herpes simplex virus type 1 (HSV-1) and 2 (HSV-2), can be treated with acyclovir (ACV) or similar nucleotide analogs, such as ganciclovir and penciclovir [12,13]. These medications are synthetic nucleoside analogs with a high affinity for the viral thymidine kinase that can block viral DNA synthesis and terminate herpesvirus infections. ACV, or the tricyclic derivative of ACV, has been previously investigated against CyHV-3 infection in vitro and in vivo [14,15,16]. ACV monophosphate (ACV-MP) at 66.67 μM can block 66% replication of CyHV-3 in CCB cells and 25% in KF-1 cells [15]. Intraperitoneal injection of 10 mg/kg ACV or 5 mg/kg cidofovir in koi can produce a 3-day mortality delay but no significant reduction in mortality at the end of infection [16]. The tricyclic derivative of ACV 6-(4-MeOPh), or T-ACV, at a concentration of 66.67 μM had lower toxicity but only produced 13–29% inhibition against CyHV-3 infection in vitro [14]. These studies indicate that ACV and related chemicals are less effective against CyHV-3 infection than HSV infection. Therefore, better antiviral therapy is needed to mitigate the disease associated with CyHV-3 infection in ornamental koi.

Replication of herpesviruses is sequentially regulated in gene transcription and translation. Transcription of the herpesvirus genome proceeds in a coordinated cascade-like fashion within the infected cells [17,18]. The immediate-early (IE) genes, such as HSV-1 ICP4 and ICP0 and CyHV-3 ORF139 and ORF15, are the first to be transcribed and are essential for early (E) and late (L) gene expression, which is required for the successful production of infectious virions [17,19]. Drugs capable of inhibiting IE gene expression will terminate herpesvirus replication early in the infection cycle and are an appealing strategy for controlling herpesvirus infection or reactivation without triggering host immune responses. Histone lysine-specific demethylases (LSD1/KDM1A) and the family of Jumonji domain-containing histone demethylases 2 (JMJD2) are two types of histone demethylases (HD) that can reverse the methylation of histone lysine residues (H3K9) in the repressive chromatin [20,21]. LSD1 can remove H3K9 mono- and di-methylation markers, while JMJD2 can remove trimethylation markers [22,23] in the repressive chromatin, also called heterochromatin. Both LSD1 and JMJD2 were recently shown to be critical for activating IE genes, such as ICP0, ICP4, and ICP27, in HSV-1 lytic infections [24,25,26]. OG-L002 is a potent inhibitor of LSD1, while ML324 is a newly discovered cell-permeable inhibitor of the JMJD2 family. Both OG-L002 and ML324 were shown to be capable of suppressing HSV-1 lytic infections and reactivation from latency in mouse trigeminal ganglion explant models [24]. Therefore, OG-L002 or ML324 can be used as chromatin-based inhibitors against herpesvirus lytic infection or reactivation. Blocking IE gene activation via inhibiting histone modification should terminate virus replication early, prevent immunogenic viral protein production, and avoid inflammation from virus infection. Drugs that can prevent viral gene activation at the early stage of virus replication will not only prevent lytic infection but also reduce the inflammatory response to viral protein products. In this study, HD inhibitors ML324 and OG-L002 were tested against CyHV-3 infection in vitro and in vivo.

## 2. Materials and Methods

### 2.1. Viruses, Cells, and Histone Demethylase Inhibitors

The United States strain of CyHV-3 (CyHV-3-U) was a gift from Dr. Ronald Hedrick. The common carp brain (CCB) cell line (gift of Dr. Scott LaPatra) and the koi fin (KF-1) cell line (a gift of Dr. Ronald Hedrick) were cultured in Dulbecco’s modified Eagle’s medium (DMEM) (Invitrogen, Carlsbad, CA, USA) supplemented with 10% fetal bovine serum (FBS) (Gemini Bio-Products, West Sacramento, CA, USA), penicillin (100 U/mL), and streptomycin (100 µg/mL) (Sigma-Aldrich, Inc. St. Louis, MO, USA) and incubated at 22 °C. CyHV-3 viral stocks were prepared by infecting CCB cells in 75 cm^2^ flasks with CyHV-3-U at 0.1 multiplicity of infection (MOI) and maintained in DMEM supplemented with 5% FBS, penicillin (100 U/mL), and streptomycin (100 µg/mL) and incubated at 22 °C. As reported previously, the viral stock was titrated by a standard plaque assay [8,27]. ML324 (cat no. S7296) and OG-L002 (cat no. S7237) were purchased from Selleckchem (Selleck Chemicals LLC, Houston, TX, USA). Acyclovir (ACV) was purchased from BioVision (Milpitas, CA, USA).

### 2.2. CyHV-3 Plaque Reduction Assay Following Drug Treatment

KF-1 cells were seeded in 12.5 cm^2^ tissue culture plates with approximately 2 × 10^5^ cells/plate on the day before infection. Before the treatment, three plates were infected with 300 µL of CyHV-3 at ~4 × 10^3^ plaque-forming units (PFU)/mL. Following a 1-h viral absorption, the cells were cultured in the presence or absence of HD inhibitors for 24 h, then washed once with phosphate-buffered saline (PBS) and cultured in 3 mL of 3% methylcellulose overlay media for 10 days at 22 °C in an incubator. The plates were then fixed in 20% methanol and stained with 1% crystal violet, and the plaques were counted [8,27]. The histone demethylase inhibitors were evaluated at similar concentrations and were effective against human simplex virus type 1 [24,25].

### 2.3. Histone Demethylase Inhibitor Cytotoxicity Assay In Vitro

Ninety-six-well plates were seeded with ~7 × 10^4^ KF-1 cells or ~4 × 10^4^ CCB cells per well and grown overnight at 22 °C. The cells were washed once with PBS and treated in 100 μL media containing ML324 or ACV at indicated concentrations at 22 °C for 24 h. The drug-treated cells were then washed once with PBS and replenished with fresh DMEM containing 5% FBS and antibiotics (as described above), and further incubated for 24 h. After incubation, cell viability was examined by using colorimetric cell viability kit III (XTT) (PromoKine, Huissen, Netherlands). Briefly, 50 μL of the XXT reaction solution was added to each well, and then the plate was incubated at 37 °C incubator for 3–4 h following the recommended protocols. Absorbance at a wavelength of 450 nm was read on a microplate fluoresce reader (BioTek, Winoosk, VT, USA) and recorded. Wells containing ML324 or ACV at each concentration in media without cells served as a blank to ensure that these drugs themselves were not registering fluorescence.

### 2.4. Viral Gene Transcription in the Presence and Absence of Histone Demethylase Inhibitor

Total RNA was extracted from infected cells using TRIzol according to the manufacturer’s instructions. The extracted RNA was then treated with DNase (Life Technologies, Carlsbad, CA, USA) and re-purified using an RNA isolation column from Qiagen (Hilden, Germany). cDNA was synthesized with the SuperScript VILO cDNA synthesis kit (Life Technologies) and used as a template for the detection of viral gene expression with primers specific for viral genes (Table 1). The selection of primers for CyHV-3 sequence amplification was based on DNA sequences of CyHV-3 (GenBank Accession Number: NC_009127.1) available at NCBI. Briefly, a 25 μL PCR reaction consisting of 12.5 μL of 2× Reaction Mix (Platinum Taq), 400 μM each of the forward and reverse primers, and about 2 μL of cDNA as template. The mixture was subjected to 94 °C for 2 min, then 40 cycles of 94 °C for 15 s, 58 °C for 30 s, and 68 °C for 45 s, followed by a 5-min extension at 68 °C. RT-PCR products were electrophoresed through a 1.5% agarose gel and then visualized by UV illumination after staining with ethidium bromide (1 μg/mL). Commercially available 1 kb Plus Ladder (Life Technologies) served as size markers. The amplification of the 18S rRNA was performed as an internal control to ensure that comparable levels of input RNA were used in a reverse-transcription-polymerase chain reaction (RT-PCR), according to the manufacturer’s instructions (QuantumRNA classic II universal 18S internal standard kit; Ambion, Life Technologies).

### 2.5. Koi Infection and Drug Treatment

Koi at 10–15 cm (6 months to 1 year old) or 5–8 cm (3–4 months old), including both males and females, were acquired from a local distributor in Oregon, quarantined for 10 days, and tested free of CyHV-3 before infection. CyHV-3 infection was carried out by immersing koi in 1× PBS containing ~1 × 10^3^ PFU/mL CyHV-3 for 30 min and then returning CyHV-3 exposed koi to tanks with one-way flow-through water and air. All treatments were given via immersion in water containing 20 µM ACV, ML324, or dilution media DMSO as vehicle control. Older koi (10–15 cm) were treated with 20 µM ML324, 20 µM ACV, or DMSO on days 1, 3, and 5 post-infection, while the younger koi (5–8 cm) were treated with 20 µM ML324, 20 µM ACV, or DMSO daily for 3–4 h on days 1–5 post-infection. Additional koi per age group served as uninfected controls. The younger koi were infected under stress temperature conditions as reported previously [8,28]. All koi were kept in 4-foot diameter tanks at the Oregon State University John L. Fryer Aquatic Animal Health Research Lab following Institutional Animal Care and Use Committee guidelines (IACUC-2021-0194).

### 2.6. Examination of CyHV-3 Infection in Body Secretions

To monitor acute CyHV-3 infection in older fish, gills and vents were swabbed with Dacron swabs on days 3, 5, and 7 post-infection. Sample swabs were placed in 0.5 mL of sterile DMEM containing 200 U/mL penicillin and 200 μg/mL streptomycin. The CyHV-3 genome copy number from each swab was estimated by real-time PCR using 10 µL of 1:1 volume of the swab solution and SideStep Lysis and Stabilization Buffer (Agilent Technologies, Santa Clara, CA) as reported previously [8].

### 2.7. Tissue Collection for Histopathology and Total DNA Isolation

Tissue samples collected from freshly euthanized fish were either fixed in 10% neutral buffered formalin for routine histopathology or stored at –80 °C prior to total DNA extraction. Total tissue DNA was isolated from approximately 100 mg of each tissue using the EZNA Tissue DNA extraction kit (Omega Bio-tec Inc, Norcross, GA, USA). Extracted total DNA was quantified using a NanoDrop Spectrophotometer (Thermo Fisher Scientific, Waltham, MA, USA). About 0.5 μg of total DNA was used for each tissue sample in real-time PCR. Fixed tissues were processed to 3 µm thick, hematoxylin and eosin-stained slides and examined by routine bright-filed microscopy.

### 2.8. CyHV-3 DNA Real-Time PCR

The standard curve was made with a DNA template amplified from CyHV-3-U DNA that was cloned into a TOPO 2.1 PCR vector (Invitrogen), as reported previously [8]. At 100 to 10^8^ copies, the prepared plasmid was used to make the standard curve for quantifying the copy numbers of CyHV-3 DNA. Based on the Ct values, the copy number of CyHV-3 DNA was calculated from the equation derived from the standard curve, y = −1.267ln(x) + 37.202, where x is the Ct value. The PCR reaction was performed as previously described [8]. The primers used in real-time PCR, CyHV-3 86F and CyHV-3 163R, and Taqman probe CyHV-3 109P, were those previously developed by Gilad et al. (2004) [4].

### 2.9. Statistical Analysis

All statistical analyses were performed using GraphPad Prism version 7 for Windows (GraphPad Software, San Diego, CA, USA). Two-way ANOVA with multiple comparison tests was used to analyze CyHV-3 genome copy numbers and treatment survival.

## 3. Results

### 3.1. Histone Demethylase Inhibitors Reduce CyHV-3 Replication In Vitro

The evolutionary history of LSD1 (KDM1) and JMJD2 (KDM4) indicates that they are relatively conserved between humans and zebrafish [29]. We hypothesize that CyHV-3 infection requires LSD1 and JMJD2 for IE gene activation and that koi LSD1 and JMJD2 are also sensitive to OG-L002 and ML324, respectively. To test this hypothesis, koi fin cells (KF-1) were infected with CyHV-3-U at ~1 × 10^3^ PFU/plate for 1 h, then the infected cells were treated with OG-L002, ML324, and ACV at 20 µM or 50 µM for 24 h, respectively. The PFU were counted on day 10 post-infection. As shown in Figure 1A, 100% inhibition of CyHV-3 plaque formation was observed in cells treated with ML324 at 20 µM or 50 µM for 24 h, and about 96% and 98% inhibition were observed in cells treated with OG-L002 at 20 µM and 50 µM, respectively. About 94% inhibition was observed in cells treated with ACV at 50 µM. These results suggest that OG-L002 and ML324 inhibited CyHV-3 replication by interfering with koi LSD1 and JMJD2 activity, respectively. Since ML324 had a higher inhibition against CyHV-3 infection in vitro, the rest of the study is focused on ML324. To determine the minimal treatment time needed for blocking CyHV-3 replication, KF-1 cells were infected similarly as above and treated with 20 µM ML324 for 10, 30, 60 min, and 24 h, respectively, immediately after CyHV-3 absorption. As shown in Figure 1B, the 10-, 30-, and 60-min treatments all significantly reduced virus replication; however, the 24 h treatment had a stronger effect than the other treatments. To find when the treatment was effective after CyHV-3 exposure, ML324 was introduced to the CyHV-3 infected cells at 1, 3, and 5 dpi, respectively. As shown in Figure 1C, ML324 treatment was only effective when the drug was given before 3 dpi. These results indicate that ML324 is only effective against CyHV-3 during the early stages of infection.

### 3.2. Cytotoxicity of Histone Demethylase Inhibitor ML324

To determine whether ML324 is cytotoxic to KF-1 cells or CCB cells, the cell viability was evaluated after exposure to ML324 at 1–100 µM for 24 h. Figure 2 shows viability results from both types of cells treated with ML324 over a concentration of 1 to 100 µM for 24 h. No significant difference in viability was apparent between mock-treated and 24 h treated cells with up to 50 µM ML324 in KF-1 cells. However, a statistically significant difference was noticeable between mock-treated and 100 µM ML324-treated KF-1 cells (*p* = 0.001). Similarly, a significant reduction in viability was also observed in 24 h treatment with 100 µM ML324 in CCB cells (*p* = 0.033). Since 20 µM ML324 is non-toxic to both KF-1 cells and CCB cells and can almost completely block 1 *×* 10^3^ PFU CyHV-3/plate infection in vitro, this concentration was selected for the in vivo study. OG-L002 and ACV cytotoxicity were measured similarly to ML324 in KF1 cells and CCB cells. No apparent cytotoxicity was observed in either KF-1 cells or CCB cells treated with OG-L002 or ACV at 1–100 µM for 24 h (Appendix A).

### 3.3. Histone Demethylase Inhibitor ML324 Blocks Transcription of CyHV-3 Immediate Early Genes

We hypothesized that ML324 could block viral genome histone demethylation and prevent CyHV-3 IE gene expression, which will prevent lytic replication. Results shown in Figure 1C demonstrate that ML324 is only effective when the treatment occurs within 72 hpi. CyHV-3 replicates slowly in KF-1 cells in vitro and takes 8–10 days to develop 80–100% CPE in the infected cells. To confirm that ML324 interferes with the transcription of viral immediate-early (IE) genes, viral IE gene expression was monitored at 6 h, 24 h, and 72 h post-infection. CyHV-3 *ORF139*, and *ORF155* were reported to be expressed within 2–4 hpi; therefore, they are considered IE genes [30]. *ORF144*, *ORF46*, and *ORF78* were unknown at this point. As shown in Figure 3, transcription of *ORF139* and *ORF155* was only detectable at 24 hpi or 72 hpi in the untreated cells but not in the ML324-treated cells. *ORF6* is a latency associate gene [31] detectable in untreated cells but not in ML324-treated cells at 72 hpi. Similarly, *ORF46*, *ORF78*, and *ORF144* were detectable at 24 and 72 hpi in untreated cells but not in the ML324-treated cells. Our studies confirmed that blocking histone demethylase with ML324 will prevent CyHV-3 IE gene expression and consequently prevent lytic CyHV-3 infection.

### 3.4. ML324 Reduced CyHV-3 Shedding in over 6-Month-Old Koi at Low Temperature

ML324 is a newly discovered small molecule. No method has been developed that could be used to determine its pharmacokinetics and half-time in a treated animal’s blood. We hypothesized that the effective in vitro dose would also be effective for in vivo treatment. In addition, we hypothesized that the drug could be delivered via immersion bath treatment (non-circulating bath). To determine whether ML324 immersion bath treatment could prevent CyHV-3 infection in vivo, 10–15 cm koi, at about 6 months of age, were infected with ~1 *×* 10^3^ PFU/mL of CyHV-3-U for 30 min in an immersion bath and then treated in an immersion bath containing either ML324 or ACV at 20 µM for 3–4 h on days 1, 3, and 5 post-infection (Figure 4A). In the control group, koi were infected similarly but only treated in an immersion bath containing the dilution solution, DMSO (Figure 4A). No death was observed in infected koi from either the treated or the untreated group. Virus shedding was monitored by gill and vent swabbing on days 3, 5, and 7. As shown in Figure 4B,C, CyHV-3 was only detectable in the DMSO control and ACV-treated groups on day 7 post-infection in the gill swabs and on days 5 and 7 in the vent swabs. However, no CyHV-3 was detected in koi treated with ML324 on days 3, 5, and 7 post-infections in the vent or gill swabs. To evaluate the histopathology in different treatment groups, tissues, including liver, kidney, spleen, intestines, gills, and heart, from three koi of each group were taken on days 3 and 7 post-infection and fixed in 10% formalin. The tissue sections from each group were examined by H&E staining. No histological difference was observed in the treated and untreated groups on day 3 post-infection. On day 7 post-infection, mild lymphocytic to histiocytic dermatitis was only seen in the DMSO-treated groups, not in the ACV or ML324-treated groups. No significant difference was observed in other tissues from the treated or untreated group. This study indicates that ML324 immersion treatment can reduce CyHV-3 replication in older koi.

### 3.5. CyHV-3 Induced Mortality Is Different at Different Temperatures in Younger Koi

Younger koi or fry are more susceptible to CyHV-3 infection [2]. High mortality from CyHV-3 infection was often reported in fry during the summertime [32]. To determine the effect of temperature on CyHV-3 infection in 3–4-month-old koi, two groups of koi, ten koi per group, were infected with ~1 *×* 10^3^ PFU/mL of CyHV-3-U for 30 min in an immersion bath. One group of koi was kept in a tank maintained at 14 °C after infection. The other group was kept in a tank with increased temperatures, as shown in Figure 5A. The tank temperature was increased at 2 °C per day from day 1 post-infection until 22 °C on day 5 post-infection; the tank temperature was maintained at 22 °C for two days, then was decreased 2 °C per day until 14 °C. As shown in Figure 5B, by day 15 post-infection, all koi died in the tank with increased temperatures, and most deaths occurred on days 5–7 post-infection. However, only 30% mortality was seen in tanks maintained at 14 °C, and the mortality was delayed until day 11 post-infection.

### 3.6. ML324 Reduced Mortality in CyHV-3 Infected Younger Koi

To determine whether an ML324 immersion bath treatment could reduce mortality in 3–4-month-old koi, three groups of koi were infected as above and then treated as depicted in Figure 6A. Following CyHV-3 exposure, the tank water temperature was increased as shown in Figure 5A. From days 1 to 5 post-infection, 20 koi per group were treated daily in an immersion bath containing 20 µM ML324, 20 µM ACV, or dilution media (DMSO) for 3–4 h, respectively. Koi are considered dead when floating without gill movement in the water or sunken to the bottom of the tank. The DMSO-treated group had ~90% mortality by day 11 post-infection, and mortality started as early as day 4 post-infection (Figure 7A). However, the ML324-treated group had only about ~30% mortality by day 11 post-infection, and mortality started two days later than in the DMSO-treated group. The ACV-treated group had about 78% mortality by day 11 post-infection, slightly lower than the DMSO-treated group, but no significant reduction in mortality at the end of the experiment (Figure 7A). ACV treatment resulted in a one-day delay in mortality compared with the DMSO-treated group. To determine if the reduced mortality was related to the reduced CyHV-3 replication in tissues, three koi from each treatment group shown in Figure 7A were euthanized on days 2, 4, and day 7 post-infection, and CyHV-3 genome copy numbers in the gills and intestines were determined by real-time PCR specific for CyHV-3. All the selected koi looked normal without erratic or irregular movements. CyHV-3 genome copy numbers were relatively low on day 2 and day 4 post-infection in both untreated and treated groups in both gills and intestines. However, the CyHV-3 genome copy number was significantly increased by day 7 post-infection (Figure 6B,C). CyHV-3 genome copy numbers in the intestines and gills from DMSO sham treatment and ACV-treated koi were significantly greater than in the intestines and gills from ML324-treated koi (Figure 6B,C). These results suggest that ML324 therapy could decrease CyHV-3 replication and reduce the mortality of koi due to CyHV-3 infection.

To determine whether the treatment effect varies in different tanks, the same treatment was repeated in three groups of koi at 10 Koi per tank. The treatment was the same as depicted in Figure 6A. Koi were maintained in tanks with elevated temperatures, as shown in Figure 5A. Figure 7B shows an average survival of 54%, 32%, and 27% at 12 dpi and 50%, 32%, and 19% at 15 dpi in the ML324, ACV, and DMSO treatment groups, respectively. Although there was no statistical difference within each treatment group, there was a significant difference between the ML324 and DMSO treatment groups. The survival rate in the ML324 treatment group was significantly higher than that in the DMSO treatment group, with a *p*-value of 0.0205. However, the ACV treatment did not significantly reduce mortality compared to the DMSO treatment. All 10 koi survived in the uninfected and untreated group.

### 3.7. ML324 Reduced Histopathology in CyHV-3 Infected Younger Koi

To determine the ML324 treatment effect against histopathology from CyHV-3 infection, three koi from each treatment group shown in Figure 7A were euthanized on days 7 and 10 post-infection and examined by H&E staining. Three normal-looking koi were selected on day 7 post-infection, and three sick or dead koi were selected on day 10 post-infection. Histopathology was evaluated in gills, intestines, kidneys, liver, and skin on days 7 and 10 post-infection in all treatment groups. Milder inflammation was observed in the gills (Figure 8A), intestines, and kidneys of ML324-treated koi compared to ACV- and DMSO-treated koi on day 7 post-infection. Gill filaments had epithelium loss (Figure 8). Moderate lymphocytic branchitis with hyperplasia and shorter filaments was seen in DMSO- and ACV-treated koi (Figure 8C, black arrows). On day 10 post-infection, all had enteritis and branchitis at a variable degree in both treated and DMSO groups, and the gill filaments were thicker and shorter, or the epithelium was lost. The DMSO sham-treated and ACV-treatment groups had more severe granulocytic dermatitis than the ML324 treatment group on day 10 post-infection. Necrosis (Figure 8D, dash outline) and intranuclear inclusion bodies (Figure 8D, green arrows) were visible in the livers of koi that died on day 10 post-infection from the ACV-treated group. These results demonstrate that ML324 immersion treatments could reduce the histopathology associated with CyHV-3 infection within 7 dpi but did not prevent all treated koi from CyHV-3 infection and tissue damage at the later stage of infection.

## 4. Discussion

Koi are valuable ornamental fish collected by many koi hobbyists throughout the world. CyHV-3 is the most pathogenic virus in koi and common carp, especially in younger koi and fry [33,34]. There are limited therapies available against CyHV-3 infection in koi. Multiple vaccine candidates against CyHV-3 infection have been investigated, including DNA, bacterial vector, inactivated, conventionally attenuated, and recombinant attenuated vaccines [35,36,37,38,39,40]. However, their protection against challenges ranges from 50% to 80%. In addition, the challenge with vaccine protection against infectious diseases in fish is that they lack sophisticated adaptive immune defense systems, the secondary lymphoid organ. Long-term protection requires specific long-term immune memory selected in the secondary lymphoid organs [41,42]. Furthermore, vaccination does not prevent the reactivation of herpesvirus latency, which was demonstrated in other herpesvirus infections, such as the pseudorabies virus [43,44]. Anti-herpesvirus drugs, such as acyclovir (ACV), acyclovir monophosphate (ACV-MP), and *Arthrospira platensis*, have been investigated against CyHV-3 and were found to have limited effects against CyHV-3 infection in vitro and in vivo [14,15,16]. One study found that ACV can delay mortality when administered intracelomically [16]. In this study, we examined both ACV and inhibitors of histone demethylase against CyHV-3 infections in vitro and in vivo.

Histone demethylases (HD) are enzymes that remove methyl (CH_3_-) groups from nucleic acids, proteins (histones), and other molecules. They belong to the lysine-specific demethylase (LSD) family or the Jumonji-domain-containing proteins (JmjC) family. They are important in transcriptional regulation, post-translation modification, and DNA damage repair [45,46,47]. OG-L002 is a potent and specific lysine-specific demethylase 1 (LSD1) inhibitor developed recently with an excellent IC50 of 20 nM [24]. LSD1 is an HD that removes methyl groups from lysine 4 or 9 of H3 histone tails, which is important in chromatin remodeling during gene activation [48]. Inhibition of LSD1 with OG-L002 was found to have inhibitory activity against *Equine herpesvirus 1*, HSV-1, and herpes zoster viral infections [26,49,50,51]. In our study, OG-L002 at 20 µM and 50 µM can produce 96% and 98% inhibition against 1–2 *×* 10^3^ PFU CyHV-3 replication in vitro, which was better than the inhibition produced by 50 µM ACV against a similar amount of HSV-1 infection in vitro [52]. We did not evaluate OG-L002 therapeutic effects against CyHV-3 in vivo since it was less effective than ML324 at a similar concentration. *ML324* is a potent JMJD2 HD inhibitor that can also suppress HSV-1 and CMV infection in vitro [24,53]. LSD1 only removes H3K9 mono and di-methylation; demethylation of the tri-methylated H3K9 is accomplished by JmjC enzymes, which represent the largest class of N ε-methyl lysine demethylases [54]. These enzymes actively participate in the regulation of gene transcription by antagonizing the action of a class of repressive histone methyltransferases, such as histone lysine methyltransferase (KMT) enzymes, H3K36 methyltransferases, and *SETD1B* [55,56]. Of the JmjC family of enzymes, JMJD2E can be inhibited explicitly by ML324. ML324 has excellent cell permeability, which may contribute to the effectiveness we observed in our in vitro and in vivo studies. ML324 has been shown to have potent antiviral activity against both HSV-1 and CMV infection via inhibition of viral IE gene expression [24,26]. In agreement with those previous reports, we also found that ML324 reduced CyHV-3 IE gene expression, such as *ORF139* and *ORF155* (Figure 3). We found that ML324 treatment was more effective at the beginning of the infection and less effective after 3 dpi (Figure 1C). This validates that ML324’s mode of action against CyHV-3 is through blocking IE gene expression at the beginning of replication. We found that treating the cell with ML324 for 10–30 min resulted in a 90% reduction of CyHV-3 infection, which demonstrates that ML 324 has excellent cell permeability (Figure 1B).

Our study found CyHV-3 infection did not cause any mortality in 6–12-month-old koi when they were kept at 14–15 °C post-infection. In addition, CyHV-3 infection only yielded about 30% mortality in 3–4-month-old koi when they were maintained at 14 °C (Figure 5B). High mortality rates (80–100%) were only observed in 3–4-month-old koi when they experienced elevated temperature to optimal temperature for virus replication within the first 5 days post-infection. In older koi, CyHV-3 infection did not produce any mortality due to low temperature, but virus shedding was reduced significantly in gills and vents in ML324-treated koi (Figure 4B,C). ML324 delivered in the water could reduce mortality by 30–60% in 3–4-month-old koi under temperature including optimal for viral replication and possible stress by temperature changes (Figure 7A,B). In addition, ML324 treatment decreased CyHV-3 replication in both gills and intestines on day 7 post-infection in 3–4-month-old koi, compared to the DMSO-and ACV-treated groups (Figure 6B,C). The ML324 treatment effect is only effective during the early infection stage. It will be interesting to see if a combination of ML324 with drugs against late-stage infection will produce better protection against CyHV-3 infection. ML324 treatment effects did vary between different treatment tanks. However, the tank effect was not statistically different within the same treatment groups (Figure 7B).

Acyclovir (ACV) is an analog of guanosine that competes with guanosine in DNA chain elongation during viral genome replication. During herpesvirus replication, ACV is first converted to acyclovir monophosphate (ACV-MP) by the viral thymidine kinase, then further converted to acyclovir triphosphate (ACV-TP) by cellular kinase. Subsequently, ACV-TP gets incorporated into the newly synthesized DNA, which terminates the DNA chain elongation during the viral genome replication. ACV is effective against HSV-1 and HSV-2 replication and is routinely given to patients with herpes keratitis or genital herpesvirus [52,57]. In this study, ACV at 50 µM produced 94% inhibition against ~1 × 10^3^ PFU CyHV-3 replication in vitro, which is comparable to the ACV effect against HSV-1 or HSV-2 in vitro (47, 48). Our study found ACV could prevent CyHV-3 infection in vitro. However, ACV is less effective against CyHV-3 infection in vivo (Figure 7A, B). The mortality in ACV treatment groups was 68–78%, which is slightly lower than the 81–90% mortality observed in the DMSO-treated group. ACV treatment produced only a 1-day mortality delay compared to the DMSO treatment. It is possible that ACV cell permeability is lower in koi or not absorbed well in a water immersion treatment. An alternative delivery, such as IP or IV, may improve the therapeutic effect of ACV against CyHV-3 infection in vivo. On the other hand, ML324 delivered in water immersion was more effective than ACV. The difference could be that ML324 has better cell permeability than ACV in water. We chose the immersion treatment in this study because this delivery method is less invasive and causes less stress to the koi. However, the immersion bath treatment method may not be the most effective drug delivery method. It will be interesting to investigate whether *intravenous* (IV) or *intraperitoneal* (IP) delivery can produce a better therapeutic effect in the future.

This study found that older koi maintained at 14–15 °C were less susceptible to CyHV-3 infections. Infection of >6-month-old koi with CyHV-3 at about 1 *×* 10^3^ PFU/mL via immersion did not cause any mortality but did produce inflammatory responses in the DMSO-treated koi. Only mild inflammatory responses were observed in ML324 and ACV treatment by day 7 post-infection. Although older koi were less susceptible to CyHV-3 infections at 15 °C, ML324 treatment did lower the rate of CyHV-3 replication in the gills and intestines at day 7 post-infection (Figure 4B,C). On the other hand, the younger koi (3–4 months old) were more susceptible to CyHV-3 infections when they experienced temperature changes. Infection of younger koi with CyHV-3 at 1 *×* 10^3^ PFU/mL resulted in 80–100% mortality in the vehicle-treated control group (Figure 7A,B). Tissue damage from CyHV-3 infection in DMSO-treated koi was similar to those reported previously [11]. Inflammatory responses were more pronounced in the DMSO- and ACV- treatment groups after day 7 post-infection. CyHV-3 DNA replication was significantly higher in gills and intestines in DMSO and ACV treatment groups (Figure 6B,C). These results suggest that mortality observed in the DMSO, and ACV treatments could have resulted from viral-induced cell damage, heat stress, and host-inflammatory responses.

ML324 had little toxicity in KF-1 and CCB cells when treated with up to 50 µM per well for 24 h (Figure 2). However, direct exposure of ML324 to KF-1 or CCB cells for 24 h had significant toxicity when the treatment concentration was increased to 100 µM. OG-L002 and ACV at 1–100 µM have no detectable toxicity to KF-1 or CCB cells (Appendix A). Since ML324 had no toxicity in KF-1 and CCB cells at 20 µM and is effective against CyHV-3 replication in vitro, the ML324 bath treatment was selected at 20 µM for 3–4 h. The pharmacokinetics of ML324 have yet to be determined since no method is available to detect ML324 in blood samples. There was no apparent histological toxicity observed in koi treated with ML324. Most of the histopathology in ML324-treated koi was related to inflammatory responses from CyHV-3 infections, which were also seen in the DMSO-treatment group. Koi that survived the ML324 treatment were kept for more than one year for a CyHV-3 latency study. No apparent physiological abnormality was observed in those treated with ML324.

## 5. Conclusions

In summary, both ACV and ML324 are effective against CyHV-3 infection in vitro. However, ACV is less effective than ML324 against CyHV-3 infection in vivo. It is possible that the ACV delivery route needs to be IV or IP to produce better protection against CyHV-3 infection in vivo. Our study demonstrated that ML324 is effective against CyHV-3 infection both in vitro and in vivo. ML324 at 20 µM delivered in water can decrease virus replication and mortality associated with CyHV-3 infection. Our study is the first to test the potential of ML324 treatment against herpesvirus infection in vivo. It is promising to find that ML324 can significantly reduce CyHV-3 replication and decrease the mortality associated with CyHV-3 infection in vivo. However, more studies are needed to find the optimal delivery route, dose, and minimal treatment time to produce optimal protection against CyHV-3 infection.

## Figures and Tables

**Figure 1 viruses-15-00163-f001:**
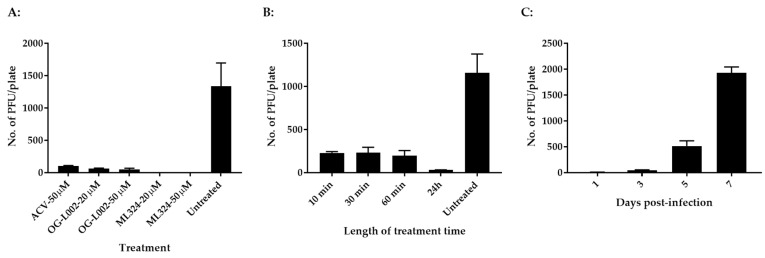
CyHV-3 plaque formation in the presence of ML324, OG-L002, or ACV. (**A**) Koi Fin (KF-1) cells treated with ACV, OG-L002, or ML324. For each treatment, three plates were first infected with CyHV-3-U at ~1 × 10^3^ PFU/plate, and then each plate was treated with ACV, OG-L002, or ML324 at 20 µM or 50 µM on day 1 post-infection for 24 h, respectively. Plaque formation was quantified 10 days post-infection (dpi). (**B**) KF-1 cells were infected similarly as above and then treated with 20 µM for 10, or 30, 60 min, and 24 h, respectively, immediately after virus absorption. After the treatment, each plate was washed once with PBS and covered with 3% methylcellulose overlay media. Plaque formation was counted on 10 dpi. (**C**) KF-1 cells were infected as above and then treated with ML324 at 20 µM for 24 h on days 1, 3, 5, and 7 post-infection, respectively. Plaque formation was quantified on 10 dpi.

**Figure 2 viruses-15-00163-f002:**
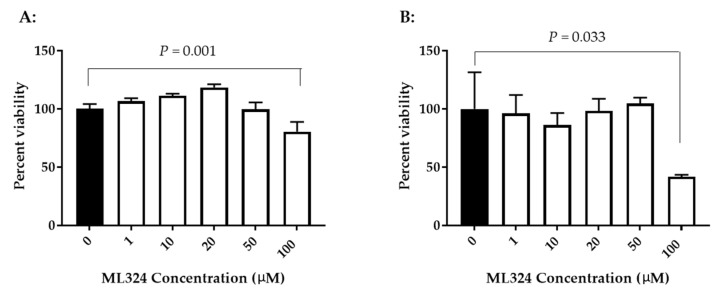
The cytotoxicity of ML324 in vitro. KF-1 cells (**A**) or CCB cells (**B**) were incubated with the indicated concentration of ML324 for 24 h. The treatment was then removed, and the cells were washed once with PBS and further incubated for 24 h in tissue culture media (DMEM) with 5% serum and antibiotics as described in the materials and methods. Cell viability was evaluated with the XTT cell viability kit III and expressed as a percentage of the mock-treated control (*n* = 3). A significant statistical difference between the mock-treated control (0) and 100 µM ML324 is marked above the line with a Bonferroni-corrected *p*-value calculated using a two-way ANOVA. The solid black column represents the mock-treated control. The open column represents the ML324 treatments.

**Figure 3 viruses-15-00163-f003:**
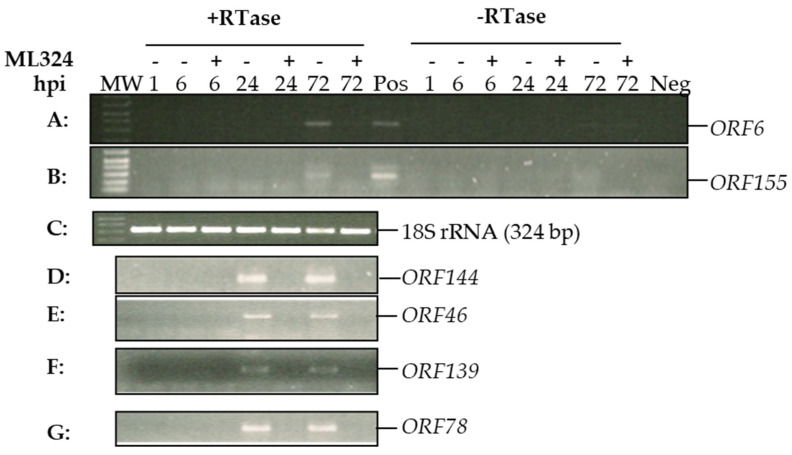
CyHV-3 viral gene transcriptions at different time points post-infection in the presence or absence of ML324. Each plate was cultured without or with ML324 at 20 µM immediately after CyHV-3-U 1 h absorption. Total RNA was isolated at 6, 24, or 72 h post-infection (hpi). Total RNA was treated with DNase before cDNA synthesis. cDNA was synthesized in the presence (+) or absence (−) of reverse transcriptase (RTase). RT-PCR products of *ORF6* (**A**), *ORF155* (**B**), *ORF144* (**D**), *ORF 46* (**E**), *ORF139* (**F**), and *ORF78* (**G**). (**C**): RT-PCR amplification of 18S rRNA showing the 324-bp product from each time point as in panels A. MW: molecular weight marker 1 kb Plus (Invitrogen), shown are 100 bp to 600 bp.

**Figure 4 viruses-15-00163-f004:**
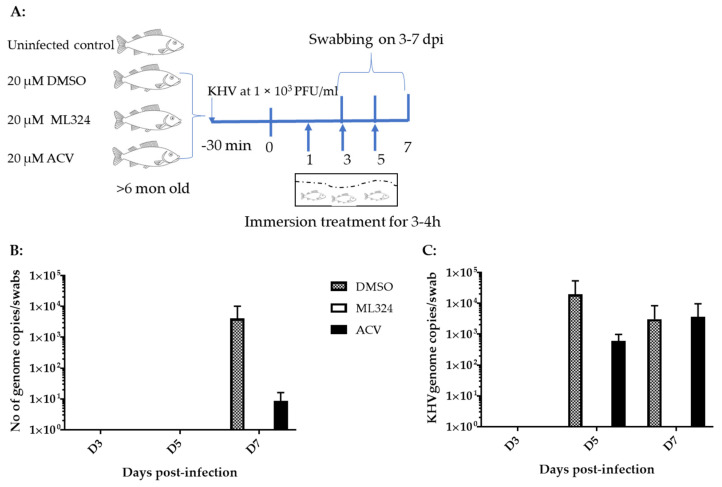
Schematic of the experimental protocol and CyHV-3 genome copy numbers in koi gill and vent swabs. (**A**) Before the immersion treatment, koi were infected via immersion bath containing 1 *×* 10^3^ PFU CyHV-3-U per ml for 30 min. Four groups of koi at 15 koi per group, at about six months to 1-year-old, were kept separately in a 3 ft tank with flow through water maintained at 15 °C. The treatments were given on days 1, 3, and 5 post-infection via immersion bath containing 20 µM ML324, ACV, or dilution media DMSO, respectively. The number represents the average CyHV-3 DNA copy number estimated by real-time PCR [8] in three gills (**B**) or three vents (**C**) swabs collected on days 3, 5, and 7 post-infection. Koi were infected with CyHV-3, as described in (**A**). Infected koi were immersed in 5 L of water containing 2 mL of DMSO, 2 mL of 50 mM ACV diluted in DMSO (20 µM in immersion bath), or 2 mL of 50 mM ML324 diluted in DMSO on days 1, 3, and 5 post-infection for 3–4 h.

**Figure 5 viruses-15-00163-f005:**
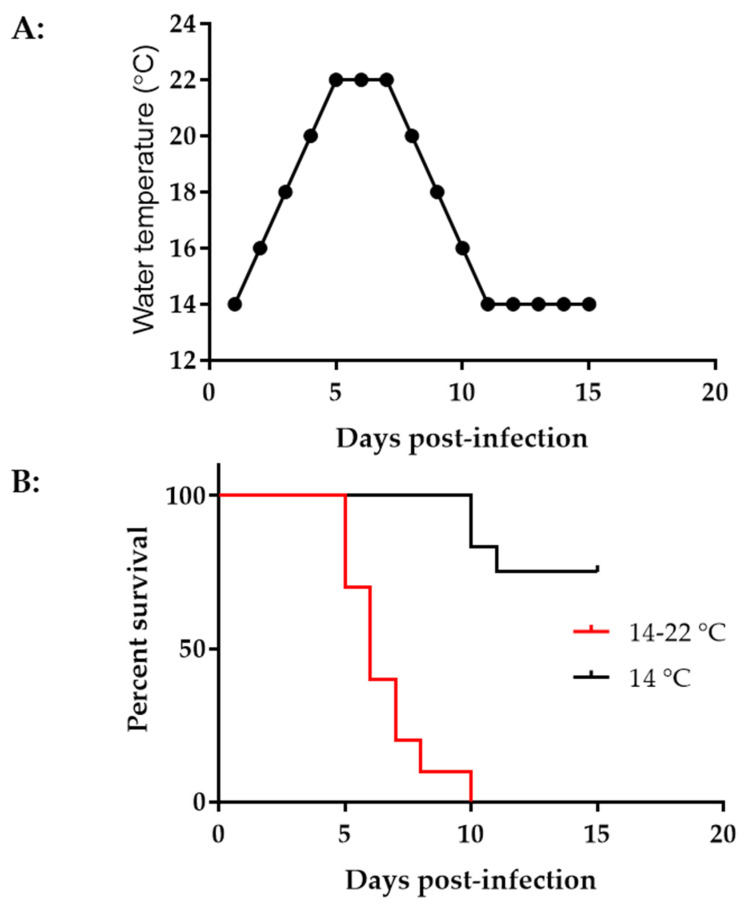
Water temperature change and mortality at different temperatures. (**A**) Tank water temperature changes on different days post-infection. (**B**) Percentage survival of koi post-infection. Koi at 3–4 months old were infected via immersion bath containing 1 *×* 10^3^ PFU CyHV-3-U per ml for 30 min and returned to tanks maintained at 14 °C (black line) or 14–22 °C (red line).

**Figure 6 viruses-15-00163-f006:**
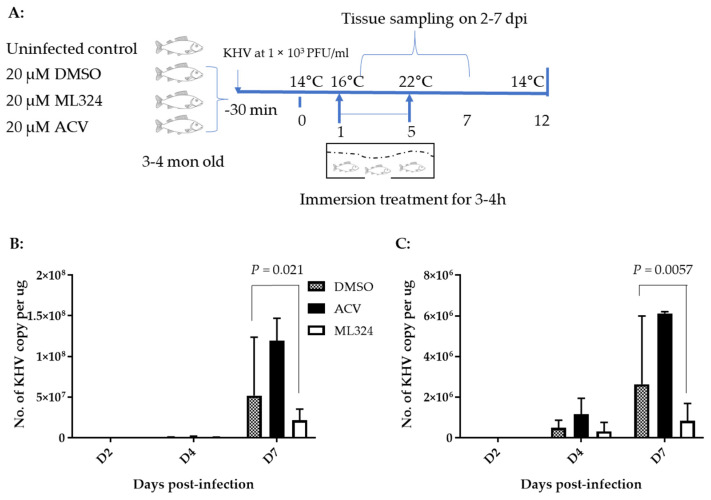
Schematic of the experimental protocol and CyHV-3 genome copy numbers in gill and vent of koi. (**A**) Four groups of koi at 20 koi per group or 30 koi with 10 koi in 3 different tanks per treatment group, at about 3–4 months old, were kept separately in a 3 ft tank with flow through water maintained at 14 °C to 22 °C during the first 5 days post-infection, then kept at 22 °C for 2 days, with the temperature lowered to 14 °C at a rate of 2 °C per day. The treatments were given daily on days 1 to 5 post-infection via immersion bath containing 20 µM ML324, ACV, or dilution media DMSO. CyHV-3 genome copy numbers were estimated in intestines (**B**) and gills (**C**) collected on days 2, 4, and 7 post-infection.

**Figure 7 viruses-15-00163-f007:**
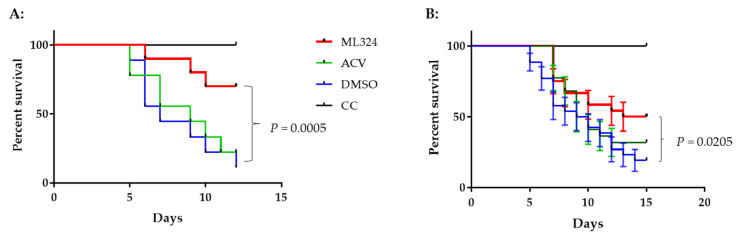
Survival of CyHV-3-infected younger koi treated with ML324, ACV, or DMSO. (**A)** Survival of koi infected with CyHV-3 in treated and untreated groups. Three to four-month-old koi in a single tank were infected with 1 × 10^3^ PFU/mL for 30 min as described in Figure 6A. (**B**). The average survival of younger koi infected with CyHV-3 in three different tanks with ML324, ACV, or DMSO. A significant statistical difference between the DMSO-treated and ML324 treated is marked with a *p*-value calculated using a two-way ANOVA.

**Figure 8 viruses-15-00163-f008:**
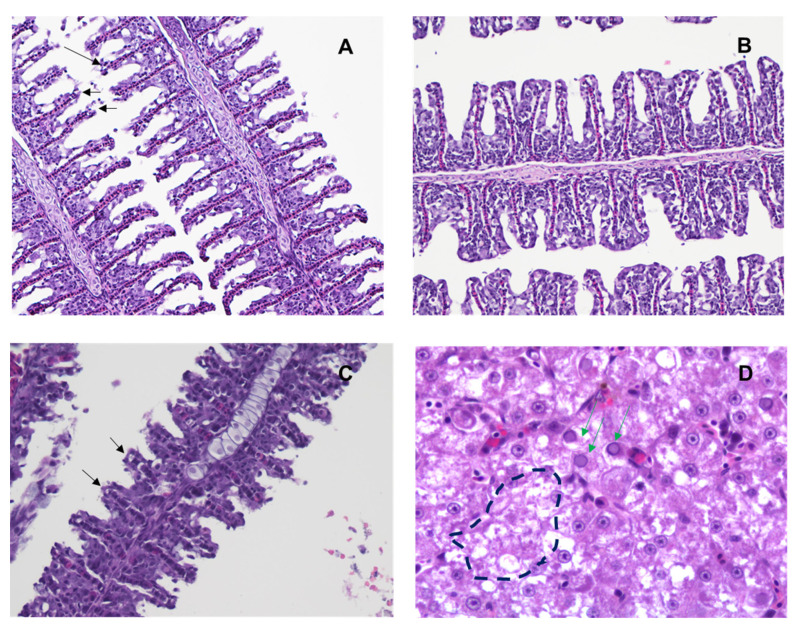
Histopathology of tissues from young koi collected on days 7 and 10 post-infection. (**A**) Gills with mild lymphocytic branchitis, sloughing of epithelial cells, and slightly shorted filament (black arrow) from the ML324-treated group. (**B**) Uninfected koi gills. (**C**) Gills with moderate lymphocytic branchitis, hyperplasia, and shorter filament (black arrow) from the DMSO-treated group. (**D**) Livers from koi that died after CyHV-3 infection from the ACV-treated group. Intranuclear inclusion bodies (green arrow) and necrosis (broken circle) were seen in liver sections. Hematoxylin and eosin (H&E) staining, 200×.

**Table 1 viruses-15-00163-t001:** Primer pairs used to detect mRNA of KHV genes post-infection.

Name	Gene	Primer Sequences (5′-3′)
	IE *	
ORF139-116-F		GCCTACTGGGAGGATATGTA
ORF139-116-R		CCCTGGTCTTGACAGAAATAG
ORF155-134-F		GGAGAGGAGAAGGGAGAAAT
ORF155-134-R		GAGTAGTTGTGGGTGATGAAG
	E **	
ORF46-103-F		GTCGATAGCGTCCTACTTTG
ORF46-103R		GACGCTCTGGTTGATGTT
ORF144-122-F		CGGTGCGACAGATACATAGA
ORF144-122-R		GATAGAGGAGAGGGTGAAGAG
	L ***	
ORF78-103-F		CCTCTGTACAACAACCCAATAA
ORF78-103-R		GTGTATTGCTGGATGGAAGG
	Latency ^	
ORF6s-F291		GACCCAGGGGACAGCTCTAT
ORF6s-R291		AGTGGTACAAGTGGCGCTTC

* Immediate early gene; ** early gene, *** late gene, ^ latency associated gene.

## Data Availability

Not applicable.

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
