# Peer review of "Evaluation of Histone Demethylase Inhibitor ML324 and Acyclovir against Cyprinid herpesvirus 3 Infection"

_viruses, 2023, doi:10.3390/v15010163_

Round 1

Reviewer 1 Report

This article aims to demonstrate the efficacy of ML324 on reduction of CyHV-3 replication in carp cells (in vitro) and carp (in vivo) post viral infection. And the results on virus load in carp cells indicates that ML324 is effective to reduce or inhibit the replication of the virus in early stage of the infection.  On the other hand, in-vivo experiments may not show clear results due to complex designs of the experiments with two sizes of fish, complicated temperature conditions and two designs of ML324 administration.   As koi carp is warm-water fish and optimal temperature for viral replications is 22-23oC, water temperature in in-vivo tests should be also fixed 22oC that is same as in in-vitro test.  

Secondary, I would suggest that experimental periods in in-vivo tests should be longer, at least 3-4 weeks after infection.  Fig.7-B and C indicate that KHV copies in fish increased between 4 dpi and 7dpi in treated group, although ML324 administration was terminated at 5 dpi. This suggest that mortality due to viral replication in treated group may continuously occur after 12 or 15 dpi.   

Third, I would recommend that additional fish for histopathology should be prepared as dead fish is inappropriate for histopathology.  It is not clear whether histopathological changes shown in Fig. 8-A, C and D are due to viral infection or change after death.  

 I am really interested in the results of Fig. 5 which suggest that ML324 treatment may be effective to inhibit KHV replication in infected fish at the low temperature where KHV can increase slowly but continuously, as these fish may become a virus carrier.  I hope that authors will continue to study on this point of view. 

Reviewer 2 Report

This paper compared two histone demethylases (HD) inhibitors OG-L002 and ML324 with acyclovir (ACV) against CyHV-3 infection in vitro and in vivo and demonstrated that ML324 has the potential to be used against CyHV-3 infection in koi, which provide us with a new idea to control CyHV-3 infection.

Overall, the article idea is good and the present study could be considered for publication after the following revisions.

1.      In Results 3.1:

1)      “SLD1” is wrong, they should be correct to “LSD1”.

2)      Line 191: Please explain the meaning of “1000–1500 PFU/plate”, the description is different with it in Materials and Methods, and we don’t know how much the “plate” represents.

3)      Line 211: The format of the figure note serial number needs to be unified, for example, all use like “(A)”. Additionally, please standardize the font format and notes of all figures in the full text and make them neatly laid out.

4)      How do you choose the concentrations of OG-L002, ML324, and ACV (20 μM or 50 μM)? Please explain and provide the references.

5)      Why didn’t you first determine the cell-safe concentrations (cytotoxicity) of the two inhibitors and acyclovir, instead of just choosing those two concentrations (20 μM or 50 μM) to try? Wouldn’t there be a suspicion that these two concentrations might kill cells? I think the first point in the results should be to determine the cell-safe concentrations of these three compounds first.

6)      Fig. 1B: Why did you set the time point of ML324 treatment in this way “10, 30, 60 min, and 24 h”? Why did it cross directly from 60 min to 24 h?

7)      Why didn’t you test for viral titers, instead of counting PFUs? How many times did you do your independent replicate experiment?

2.      In Results 3.2:

1)      Lack of OG-L002 and ACV cytotoxicity assay results.

2)      Please explain why choose “1–100 μM” to program cytotoxicity assay.

3)      Line 232–234: We don’t understand why you stress the two time points “for 4 h or 24 h”. And why don’t you provide the data of “KF-1 cells or CCB cells treated with ACV at 1-100 μM for 4 h or 24 h”?

3.      In Results 3.3:

1)      Fig.3: The indicator lines of the genes to which the strips belong should be aligned.

2)      Why didn’t you detect the effect of inhibitors on gene expression levels by real-time PCR?

3)      Why did you set the “-RTase group” only for ORF6 and ORF155 except other genes including 18S rRNA?

4)      Fig.3C: In addition to “324 bp”, it should be marked 18S rRNA.

5)      Table 1: What is “ORF6s”? Please verify them.

4.      In Results 3.4:

You should add the histopathology graphs of over 6-month-old koi.

5.      In Results 3.5:

Title: “CyHV-3-induced mortality is different at different temperatures” should be modified to “CyHV-3-induced mortality is different at different temperatures in younger koi”, otherwise readers may be misled that this part is not linked to the topic of the article, however it is actually to test the CyHV-3 infection temperature on younger koi for Results 3.6.

6.      In Results 3.6:

Fig. 4A and 4B should be divided in two parts. Fig. 4B should be placed in Results 3.6.

Writing issues:

1.      The article format needs to be revised in strict accordance with the requirements by viruses (please refer to “MDPI Style Guide Second edition” to modify them, website: https://www.mdpi.com/authors/layout#_bookmark25). Please revise the relevant format to be uniform throughout.

1)      Line 37: “-” in “2-3” should use en dash “–” to show number range, also like line 39 “[1-3]” and line 58 “13-29%”, etc. This type of issue requires a full text revision.

2)      Line 96: “22℃” (there should have a space between the number and unit) should contain a space and please unify them in full text.

3)      Line 111: In “~7 × 104” or “~4X104”, etc. “X” should use “×” and it should contain a space. Please standardize and unify the format in full text.

4)      Line 131: “ul” should be “μl” and modify them in full text.

5)      Line 319: “two ℃” should be “2 ℃”.

2.      Line 66: gene name should use italics and unify them in full text.

3.      Line 72: “JMJD2” rather than “JMID2”, please modify it.

4.      Line 83: In “Blocking IE gene activation via blocking histone modification…”, please don’t use “blocking” repeatedly and replace the synonyms.

5.      Line 92: Absence of the indefinite article “a”, and please add it.

6.      Line 94 and other similar issues in full text: In Materials and Methods, the first mention of the reagent should be followed by its company name, city, region or state and country. The second mention in the following does not need to be repeated annotation. In addition, in Line 95, the “Sigma-Aldrich, Inc.” expression seems to be not right, and there are similar issues in the following text, I do not list them one by one, please verify and modify them.

7.      Line 97–99 and other similar issues in full text: Proper nouns, for example, “Dulbecco’s modified Eagle’s medium (DMEM)” need to write the full name followed by its abbreviation when first mention, but do not need to repeat the full name on the second mention. Meanwhile, “MOI” first mention should annote its full name (multiplicity of infection).

8.      Line 105: “PFU” first mention should annote its full name (plaque-forming unit), but “plaque-forming units (PFU)” in line 193 don’t need annote full name repeatedly.

9.      Line 106 and 113: Unspecified the source of reagents for HD inhibitors (OG-L002 and ML324) and acyclovir, please add their company name, city, region or state and country.

10.  Line 130: It should be “GenBank Accession Number” rather than “NCBI accession no.”.

11.  Line 133: First mention of “minutes”, it should use full name, but the following mention should use “min”, like line 134 (or unify them in full text by “min”). Please check them in full text.

12.  Line 135: Please add the full name for “RT-PCR”.

13.  Line 171: I think the description “CyHV-3 DNA real-time PCR” is not proper, it should be modified to “Detection of CyHV-3 copy numbers by real-time PCR”.

14.  Line 180: “Gilad et al.” should be “Gilad et al. (2004)”.

Reviewer 3 Report

The manuscript is clearly presented and well-written. It reports on the antiviral therapy against CyHV-3 of ML324, a specific inhibitor of LSD1 and JMJD2 and demonstrates that ML324 could decrease CyHV-3 replication and reduce the mortality of Koi due to the CyHV-3 infection. Moreover,ML324 were compared with acyclovir (ACV) against CyHV-3 infection in vitro and in vivo and showed the better effectiveness.

  In general, it is a clear and technically well-conducted study.

   Just a question need to be discussed. Since the outcome of infection is very dependent on environmental temperatures, and as CyHV-3 is concerned, the mortality occurs when water temperatures range from approximately 18 to 25ºC. In Results 3.4. ML324 reduced CyHV-3 shedding in over 6-month-old koi, the water is maintained at 15ºC (Figure 4A).  with losses tending to occur when water temperatures range from 18-25ºC, why not choose a more susceptable temperature,such as 22ºC?

Round 2

Reviewer 1 Report

Hedrick et al (2000) demonstrated that 2 years old carp (0.8 kg in body weight) exposed to KHV and kept in 23oC showed 80% mortality in 13-17 days post viral exposure due to KHV infection.  This indicates that 1) even older carp die with KHV infection, 2) carp die with KHV infection under a constant temperature if which is optimal for viral replication, and 3) two or three weeks are necessary to observe cumulative mortality with KHV infection.

Based on these, I recommended authors to perform in vivo tests at a constant and optimal temperature for viral replication during longer period. 

However, if the authors need to publish with results in the present manuscript, I think that several descriptions in the present manuscript should be reconsidered.

In addition, I believe that carp Cyprinus carpio is warm water fish and 22-23oC is included in physiological temperature for carp.

 My opinions is as follows:

Line 270: “ML324 reduced CyHV-3 shedding in over 6-month-old koi” should be “ML324 reduced CyHV-3 shedding in over 6-month-old koi at low temperature”

 Reasons: The experiment conducted only at low temperature which is not optimal for viral replication.

 Line 281: “all koi recovered after the infection” should be removed.

 Reasons: These is no description that proves the recovery.

 Line 336: “at the end of infection” should change to be “at the end of experiment”.

 Reasons: To clear the meaning.

 Line 338-339:  The sentence should change to be “---in tissues, xx (number of fish) koi from each treatment group shown in Fig. 7A were euthanized on days 2, 4, and day 7 post-infection, and CyHV-3 genome copy numbers in the gills and intestines were determined by real-time PCR specific for CyHV-3”.

In addition, how to distinguish between fish for mortality monitoring and for sampling in group shown in Fig.7A should be described.

 Reasons: To clear the method for experiment.

 Line 376-: The sub-heading (3.7) and Fig.8 are mismatching. Histopathology of ML324 treated fish should be shown in Fig. 8.

 Reasons: Fig.8. in the present manuscript just shows histopathological findings of CyHV-3 infection in ACV- and DMSO-treated fish, which cannot support the sub-heading.

 Line 454: “high temperatures (heat stress)” should change to be “elevated temperature to optimal temperature for virus replication”.

 Reasons:  It is a fact that 22oC but not 14oC is optimal temperature for viral replication. 

 Line 455:  Should be add “due to low temperature” after “ ---did not produce any mortality”.

 Reasons:  The reason of lack of mortality is due to low temperature (not optimal temperature for viral replication).

 Line 457: “in 3-4 month-old koi under heat stress” should be “in 3-4 month-old koi under temperature including optimal for viral replication and possible stress by temperature changing.

 Reasons:  It is not obvious whether a change of temperature form 14oC to 22oC or from 22oC to 14oC induces CyHV-3 infection, although it is well known that 22oC but not 14oC is optimal for viral replication.

 Line 492-493:  Should be added “at 15oC” after ----CyHV-3 infections,

 Reasons: If the temperature is not fixed at 15oC, the sentence conflicts of the description by Hedrick et al (2000). 

 Line 496:  “temperature stress” should be “temperature changing”.

 Line 496:  “Infection of stressed younger koi” should be “Infection of younger koi”.

 Reasons: Temperature changing and temperature stress are not equal.

 Line 521: Remove “highly”.

 Reasons: Based on the results of both mortalities and viral genome copies, efficacy of ML324 administration seems to be limited.

 Line 523:  Remove “in younger fry”.

 Reasons: Decrease of viral replication in ML324 treated fish was also observed in 6 month old fish. 

Author Response

Please see the responses in the attachment. All the corrections are made in blue in the newly revised manuscript.
